# Lysine Methylation-Dependent Proteolysis by the Malignant Brain Tumor (MBT) Domain Proteins

**DOI:** 10.3390/ijms25042248

**Published:** 2024-02-13

**Authors:** Hong Sun, Hui Zhang

**Affiliations:** Department of Chemistry and Biochemistry, Nevada Institute of Personalized Medicine, University of Nevada, Las Vegas, 4505 South Maryland Parkway, P.O. Box 454003, Las Vegas, NV 89154-4003, USA; hong.sun@unlv.edu

**Keywords:** lysine methylation, SET7/SETD7, LSD1/Kdm1a, demethylation, L3MBTL3, CRL4, DCAF5, ubiquitin E3 ligase

## Abstract

Lysine methylation is a major post-translational protein modification that occurs in both histones and non-histone proteins. Emerging studies show that the methylated lysine residues in non-histone proteins provide a proteolytic signal for ubiquitin-dependent proteolysis. The SET7 (SETD7) methyltransferase specifically transfers a methyl group from S-Adenosyl methionine to a specific lysine residue located in a methylation degron motif of a protein substrate to mark the methylated protein for ubiquitin-dependent proteolysis. LSD1 (Kdm1a) serves as a demethylase to dynamically remove the methyl group from the modified protein. The methylated lysine residue is specifically recognized by L3MBTL3, a methyl-lysine reader that contains the malignant brain tumor domain, to target the methylated proteins for proteolysis by the CRL4^DCAF5^ ubiquitin ligase complex. The methylated lysine residues are also recognized by PHF20L1 to protect the methylated proteins from proteolysis. The lysine methylation-mediated proteolysis regulates embryonic development, maintains pluripotency and self-renewal of embryonic stem cells and other stem cells such as neural stem cells and hematopoietic stem cells, and controls other biological processes. Dysregulation of the lysine methylation-dependent proteolysis is associated with various diseases, including cancers. Characterization of lysine methylation should reveal novel insights into how development and related diseases are regulated.

## 1. Introduction

The lysine residues in proteins can be mono-, di-, and tri-methylated by specific methyltransferases using S-Adenosyl methionine as the methyl donor [1,2]. Lysine-specific methylation is a major protein modification, and extensive research has established the key roles of various methylated lysine residues at the amino terminal regions of histones in modulating chromatin structure and gene expression [3,4]. Various mono-, di-, and trimethylated lysine residues in histones provide specific protein recognition modules that are “read” by distinct methylation binding proteins, including proteins containing the Tudor, Chromo, or PHD domains, to regulate chromatin structure and gene expression through their specific interactions with transcription factors or chromatin modulators [3,5,6,7].

Many non-histone proteins have been found to be methylated at specific lysine residues to regulate their activity or protein stability [2,8]. For example, p53, DNA (cytosine-5)-methyltransferase 1 (DNMT1), SRY-Box2 (SOX2), E2F1, LIN28A, GLI3, Nuclear Factor Kappa B/REL-associated protein (NFκB/RelA), FOXO3, signal transducer and activator of transcription 3 (STAT3), and hypoxia-inducible factor 1α (HIF1α are methylated by SET7 (also called KMT7, SETD7, SET9, or SET7/9) [2,8,9,10,11,12], a methyltransferase that was originally isolated based on its ability to monomethylate the lysine 4 residue in histone H3 (H3K4). These proteins contain a lysine residue located in a methylation motif that often bears similarity to that of H3K4. These studies indicate that SET7 can methylate both histone H3 and non-histone proteins. Although various methylated forms of H3K4 are associated with active chromatin, the function and regulation of methylated lysine residues in non-histone proteins require further characterization.

## 2. Non-Histone Proteins Are Targeted for Proteolysis

Although various lysine residues in histones can be methylated, there is, so far, no evidence that lysine methylation induces the proteolysis of histone proteins [2,3]. However, accumulating evidence indicates that methylation of lysine residues by SET7 on a group of non-histone proteins, such as DNMT1, SOX2, E2F1, NFκB/RelA, FOXO3, and STAT3, triggers the proteolytic degradation of these modified proteins [13,14,15,16,17]. DNMT1 is a major DNA methyltransferase that transfers the methyl group to the cytosine residues in the CpG dinucleotides of the genome [18,19]. It maintains the CpG DNA methylation patterns in the newly synthesized DNA strands during semi-conservative DNA replication to preserve epigenetic inheritance [18]. During DNA replication, unmethylated cytosine residues are initially incorporated into the newly synthesized daughter DNA strand, generating hemi-methylated DNA that retains the methylation information of the parent DNA strand. DNMT1 is recruited to the replication fork to help complete the full methylation pattern through its interaction with UHRF1 (ubiquitin-like, containing PHD and RING finger domains, 1) [18,20], a protein that specifically and directly interacts with hemi-methylated DNA. The selective methylation of CG islands (CGIs) at the transcriptional regulatory regions provides a critical repressive mechanism to regulate cell- or tissue-specific gene expression [20]. Loss of DNMT1 impairs embryonic development and organogenesis in mice and other animals [20,21,22]. Aberrant methylation patterns of CGIs are associated with various human cancers and developmental abnormalities [20,22]. DNMT1 also regulates genomic imprinting, heterochromatin structure, and inactivation of the X-chromosome [19,23]. In mouse embryonic stem cells (ESCs), genetic deletion of the *Lsd1* (*Lysine-specific histone demethylase 1A,* also called *Kdm1a* or *Aof2*) alleles, which encode a demethylase that is commonly known to remove the methyl groups from methylated histone H3 at lysine 4 (H3K4) [3,24], triggers the proteolysis of the mouse DNMT1 protein and, consequently, reduces total methylcytosine by 43% in DNA, producing hypomethylation of genomic DNA [25]. It was found that SET7 [17,25] can mono-methylate mouse DNMT1 protein on lysine 1096 in vitro, whereas LSD1 can remove this methyl group from the methylated mouse DNMT1 protein. Loss of LSD1 caused the accumulation of the methylated DNMT1 and led to the proteolysis of DNMT1 protein through the ubiquitin-proteosome system (UPS) [17,25]. Several other studies have revealed that the DNMT1 protein is destabilized by specific monomethylation on lysine 142 (K142) (Figure 1A) by SET7 to trigger DNMT1 proteolysis. This methylation motif of K142 in DNMT1 overlaps with the AKT1 phosphorylation motif, and phosphorylation of serine 143 (S143) blocks the methylation of K142 in DNMT1, leading to the stabilization of the DNMT1 protein. While ubiquitin E3 ligases that specifically interact with phosphorylated, acetylated, or proline-hydroxylated protein substrates for ubiquitin-dependent degradation are well established [26,27,28,29], lysine methylation-mediated protein degradation is poorly understood.

SOX2 is a master stem cell protein that is essential for the maintenance of pluripotency and self-renewal of the ESCs, as well as the development of the inner cell mass (ICM) and extraembryonic ectoderm of blastocysts in early mouse embryos [30]. SOX2 is one of the key factors that can convert somatic cells to induced pluripotent stem cells (iPSCs) [31]. SOX2 regulates development in a dose-dependent manner [32,33,34]: even at the 4-cell embryonic stage, heterogeneous binding of SOX2 to the target genes determines the first lineage decision [35,36,37]. In ESCs, SOX2 forms a binary complex with OCT4 to activate the core transcriptional regulatory circuitry to regulate the transcription of developmentally important genes, such as *Oct4*, *Sox2*, *Nanog*, *Fgf4*, *Utf1*, and *Lefty1*, in an autoregulatory and feed-forward loop [38,39,40,41]. In fetal development, the dose-dependent expression of SOX2 is further required for many fetal endodermal and ectodermal tissues, such as the central nervous system (CNS), lens epithelium, and anterior foregut endoderm [30,42]. Mutations of one copy of the human *Sox2* gene alleles cause familial anophthalmia/microphthalmia (no eye or small eye), and individuals with these mutations may also have seizures, brain abnormalities, slow growth, delayed motor skills, and mild to severe learning disabilities [43,44,45]. Individuals with *Sox2* haplo-insufficiency are often born with esophageal atresia (a blocked esophagus), an abnormal connection between the esophagus and trachea, and male genital abnormalities, including micropenis [44]. SOX2 is also critical in progenitor cells of many adult tissues such as the brain and retina; other epithelial tissues, including the testes, forestomach, glandular stomach, anus, cervix, esophagus, and lens; as well as the glands associated with the oral cavity, trachea, and cervix [42]. Conversely, constitutive expression of mouse SOX2 in neural stem/progenitor cells within the ventricular zone inhibits neurogenesis [46]. Elevation of SOX2 levels is also associated with human diseases, particularly in malignancy. *Sox2* acts as a major oncogene that is recurrently amplified at 3q26.33 in squamous cell carcinomas of the lung, esophagus, and oral cavity [47,48,49,50,51], as well as small cell lung carcinomas and glioblastoma multiforme [52,53]. SOX2 is over-expressed in many poorly differentiated and aggressive cancers [54], including breast, ovarian, gastric, and colon carcinomas [51,55,56,57,58,59,60,61,62,63,64]. SOX2-expressing cancer cells usually act as cancer progenitor/stem cells that are resistant to various cancer therapeutic compounds [65]. In mice, SOX2-expressing medulloblastoma cells behave as the most primitive progenitor cells that promote hierarchical tumor growth [66]. Thus, the levels of SOX2 should be precisely regulated in development, and altered expression of SOX2 can cause various diseases, including cancers and developmental disorders. Indeed, early studies revealed that the mouse SOX2 protein contains a potential lysine methylation motif at lysine 119 (K119, equivalent to K117 in human SOX2)(Figure 1A) that shares similarity to that of K142 in DNMT1. K119 in mouse SOX2 can be methylated by SET7 to inhibit SOX2 transcriptional activity and induce SOX2 polyubiquitination and proteolytic degradation [67]. Similar to DNMT1, this methylation K119 motif overlaps with the AKT1 phosphorylation motif, and phosphorylation of threonine 118 blocks the methylation of K119 in SOX2, leading to the stabilization of the SOX2 protein. The report further indicated that the methylated K119 is recognized by ubiquitin E3 ligase WWP2 to regulate the proteolysis of SOX2 [67].

The lysine 185 (K185) residue in E2F1 is located in a methylation motif shared by H3K4 and DNMT1 (Figure 1A) [2,14]. E2F1 acts as a critical transcription factor regulated by the retinoblastoma tumor suppressor family proteins for the cell-cycle-dependent expression of many S-phase genes in the cell cycle. SET7 was shown to methylate K185, consequently triggering the methylation-dependent proteolysis of E2F1 [14]. The methyl group on K185 of E2F1 is dynamically removed by LSD1. While the loss of LSD1 caused increased levels of H3K4me1/me2, the inactivation of LSD1 in mouse ESCs, human teratocarcinoma PA1 cells, human colorectal carcinoma HCT116, and lung carcinoma H1299 cells led to the proteolytic downregulation of the methylated DNMT1, SOX2, and E2F1 proteins. How these proteolytic processes are mediated by ubiquitin E3 ligases remains largely unclear.

## 3. The Malignant Brain Tumor (MBT) Domain Serves as a Methyl Lysine Recognition Domain

The Drosophila *lethal(3) malignant brain tumor* gene, *l(3)mbt* [68,69], was initially identified as a tumor suppressor gene whose mutation causes malignant growth of the adult optic neuroblasts and ganglion mother cells in the larval brain, promoting the formation of a malignant, highly invasive, and lethal brain tumor and an overgrown imaginal disc. The encoded L(3)mbt protein contains a novel malignant brain tumor (MBT) domain with three tandemly repeated MBT motifs at the amino terminal region, followed by a unique zinc finger motif (zf-HC/zf-C2) and the SPM/SAM (SCM, PH, and MBT homology/Sterile α motif) domain at the middle and carboxy terminal regions [68] (Figure 2). The SPM/SAM domain is a large protein interaction module present in a wide variety of proteins to form homo- or hetero-oligomers [70,71]. The MBT domain is also found in another Drosophila protein encoded by the Sex Comb on midleg (*Scm*) gene, a Polycomb group (PcG) gene to control specific homeotic gene expression during development [71]. Mutations of *Scm* cause many homeotic genes to be mis-expressed in the central nervous system during development [71,72]. The *Scm* mutation produces the strongest PcG phenotypes, and the SCM protein serves as a central component of the PcG repression complex to regulate homeotic gene transcription. The SCM protein contains two tandemly repeated MBT motifs and the SPM/SAM domain, arranged similarly to that of the L(3)mbt protein [71]. The MBT domain was further identified in the Drosophila SFMBT protein, a component of the Pho-repressive complex (PhoRC) [73]. Similarly to L(3)mbt and SCM proteins, SFMBT protein contains four tandemly repeated MBT motifs and a conserved SPM/SAM domain (Figure 2).

The human MBT family has nine identifiable members [74,75,76,77,78,79,80,81,82,83] (Figure 2): L3MBTL1 (also called L3MBTL in humans) [74], L3MBTL2 (also called M4MBT in mice) [75], L3MBTL3 (also called MBT-1 in mice) [76], L3MBTL4 [82,84], MBTD1 (also called Hemp), SFMBT1, SFMBT2 [79,80,85], SCMH1, and SCML2 [77,78]. Among them, L3MBTL2 and MBTD1 do not contain the SPM/SAM domain. Furthermore, PHF20L1, a human protein containing a hybrid MBT domain, with an amino-half similar to the MBT motif and an adjacent carboxy-half resembling the Tudor domain, was also found [86,87,88] (Figure 2). The presence of a larger MBT protein family suggests that the MBT proteins may regulate diverse biological processes in humans and related mammals.

The MBT domain is structurally related to the Tudor, Chromo, and PWWP domains, which belong to the “Royal” family of methylation-binding protein domains [89,90]. Since lysine residues in histones can be mono-, di-, and tri-methylated [1], many early studies aimed to examine whether the MBT domains serve as a methylation reader of methylated lysine residues in histones. Various experiments, including pulling-down analysis, peptide-array screens, isothermal titration calorimetry (ICT), and fluorescence polarization, suggested that the MBT-repeated motifs preferentially bind to lower methylated states of lysine residues, such as the mono- or di-methylated lysine residues in the core histone tails, whereas these MBT motifs exhibit much lower affinity towards unmethylated or tri-methylated cognate lysine residues [5,89]. The MBT proteins usually contain several repeated MBT motifs. Structural studies reveal that only one of the MBT motifs can recognize the methylated lysine residue of the target proteins. Other MBT motifs are required to participate in the head-to-tail interdigitated interactions between the tandem MBT repeats to stabilize a propeller leaf-like structure in the MBT domain [91,92,93]. Mutational analysis of individual MBT motifs in the MBT proteins supports that only one MBT motif of a particular MBT protein is capable of binding to a methylated lysine residue. Although some MBT proteins can weakly bind to methylated lysine residues in histones, the roles of these MBT proteins remain unclear [5].

Some MBT domain proteins have been characterized previously. For example, the three MBT repeated domains (3MBT) of human L3MBTL1 were shown to preferentially interact to the mono- and di-methylated H4K20 (H4K20me1/2) and H1bK26 [H1bK26me1/2) via in vitro pull-down assays [89,92,94]. Mass spectrometry analysis revealed that human L3MBTL1 is associated with core histones, H1b, HP1γ, and retinoblastoma tumor suppressor protein pRb [94]. The MBT-repeated motifs of L3MBTL1 can compact histones into nucleosomal arrays in vitro. This process is dependent on the presence of H4K20me1/2 and H1bK26me1/2 [89]. The L3MBTL1-dependent nucleosomal compaction is due to its ability to bind to two nucleosomes simultaneously to repress a subset of E2F-regulated gene expression [94]. The p53 tumor suppressor protein is the first non-histone protein found to be specifically mono-methylated at lysine 372 (K372) by SET7 [12]. The mono-methylated K372 in p53 can be demethylated by LSD1 [95]. The lysine 382 (K382) motif of p53 was also found to resemble a consensus H4K20 methylation motif. K382 in L3MBTL1 is mono-methylated by SET8/PR-SET7 methyltransferase [96], originally identified as a H4K20 mono-methyltransferase. The mono-methylated K382 (K382me1) of p53 is read by the second MBT repeat (MBT2) in the 3 repeated MBT domain of L3MBTL1. The binding of L3MBTL1 to p53 via K382me1 can negatively regulate p53 transactivation to repress the expression of p53-target genes [97]. The MBT domain of human L3MBTL2 contains four MBT repeats, but without the C-terminal SPM/SAM domain (Figure 2) [75,98]. In vitro ICT assays and crystal structural analyses show that the fourth MBT repeat preferentially binds to the H4K20me1 peptide, similarly to that of MBTD1, another human MBT protein with the four-MBT-repeated domain [98,99]. The human *L3MBTL2* gene is recurrently deleted in patients with medulloblastomas [100]. The in vitro ICT and crystal structure analyses revealed that the fourth MBT repeat preferentially bound to the H4K20me1 peptide, similarly to that of human MBTD1, another MBT protein with four MBT repeats (Figure 2) [98,99]. Human L3MBTL2 protein was isolated as an integral component of Polycomb Repressive Complex 1-like 1.6 complex (PRC1.6) [101], along with other PRC1 proteins, including PCGF6, MGA/MAX, RING1, RING2, E2F6/DP1, RYBP/YAF2, and HP1γ, by mono-ubiquitinating lysine 119 (K119) in histone H2A (H2AK119) to repress gene expression [102,103]. In addition, the MBT domain of L3MBTL2 can induce in vitro chromatin compaction, without histone modifications, to repress gene expression [101]. While mice with homozygous deletion of mouse *L3mbtl1* alleles survive, mouse knockout of *L3mbtl2* is lethal in the early embryonic stage, even though the implantation, formation of trophectoderm, primitive endoderm, and the inner cell mass are relatively normal [104]. However, the *L3mbtl2* KO embryos cause growth retardation and developmental defects on embryonic day 6.5–7.5 (E6.5–E7.5), with the inner cell mass failing to form a normal primitive ectoderm capable of gastrulation. Deletion of *L3mbtl2* in mouse embryonic stem cells (mESCs) causes proliferation defects without affecting the pluripotency of mESCs [104]. The *L3mbtl2* mESCs grow very slowly by extending the doubling time of embryonic stem cells from about 13 h to about 33 h. In vitro, the second MBT repeat of SCML2 protein binds to mono- and di-methylated H4K20 and H3K9 peptides with dissociation constant at about 0.5–1.2 mM by ITC [105]. The MBT domains of both SCML2 and SCMH1 contain two MBT repeats (Figure 2). The affinity of the MBT2 of SCML2 to various low methylated histone peptides such as H4K20, H3K9, H3K4, H3K27, and H3K36 is relatively low, within twofold differences among these methylated peptides [5]. There is no clear preference for the binding of SCML2 to these methylated histone peptides. *Scmh1* null mice are viable and develop normally; but about half of *Scmh1* KO male mutants have smaller testes that contain fewer spermatocytes due to the high rate of apoptosis of spermatocytes [106]. Both mouse and human SCML2 proteins are encoded by an X-linked gene. Mouse *Scml2* deletion in male (*Scml2*^−/*Y*^) and female (*Scml2*^−/−^) mice is relatively normal, but the male mice have lower sperm counts and fewer littermates [107].

## 4. The Methylated DNMT1 as a Substrate for L3MBTL3 and the CRL4^DCAF5^ Ubiquitin E3 Ligase Complex

Although the roles of most MBT proteins in ubiquitin-dependent proteolysis remain unclear, recent studies have revealed that one of the MBT proteins, L3MBTL3, is involved in the lysine methylation-dependent proteolysis mediated by SET7 and LSD1 [108,109,110]. The first set of these findings started with the regulation of DNMT1 [108]. Several reports have shown that K1094 (equivalent to K1096 of mouse DNMT1) and K142 of human DNMT1 are methylated by SET7 [25,111]. It was essential to determine which lysine residue is most critical for the regulation of DNMT1 proteolysis. Using a specific LSD1 inhibitor, CBB3001, it was found that in human colorectal carcinoma HCT116 cells, inhibition of LSD1 was sufficient to induce the proteolytic degradation of DNMT1 [108]. Mutational conversion of K142 of DNMT1 to alanine (K142A) led to the resistance of DNMT1 to LSD1 inhibitors or siRNA-mediated LSD1 ablation. However, the conversion of K1094 to the arginine (K1094R) mutation in human DNMT1 is still sensitive to the loss of LSD1 [108]. In addition, recombinant LSD1 can effectively remove the methyl group from the mono-methylated K142 of DNMT1. These studies indicate that LSD1 efficiently demethylates the methylated K142 of DNMT1 and that the inactivation of LSD1 causes elevated levels of the methylated K142 to promote DNMT1 proteolysis.

The characterization of DNMT1 proteolysis was initially based on the effect of MLN4924, an inhibitor of the neddylation of cullin-RING ubiquitin ligases (CRLs) [108]. Treatment of cells with MLN4924 was found to stabilize the DNMT1 protein in LSD1 knockdown cells, suggesting that one of CRLs might serve as the ubiquitin E3 ligase to target DNMT1 for proteolysis. Since CRLs comprise the largest families of ubiquitin E3 ligases in mammalian genomes [27], siRNA-mediated knockdown analyses were initially used to examine whether one of the CRLs was involved in the methylation-dependent DNMT1 proteolysis [108]. Previous studies have revealed that the CRL1 ubiquitin ligases recognize phosphorylated protein substrates, whereas CRL2 ubiquitin E3 ligases use elongin B/C as the bridge to bind to the substrate-specific subunits, such as the SOCS proteins, to interact with the substrates [27]. One example is that the CRL2/elongin B/C employs the von Hippel–Lindau (VHL) tumor suppressor protein as the substrate binding subunit to interact with proline-hydroxylated protein substrates for proteolysis [27,28]. However, the mechanism by which methylated lysine is recognized for proteolysis remains unclear [112]. It was found that the proteolysis of DNMT1 induced by the loss of LSD1 is regulated by the cullin-RING ubiquitin ligase 4 (CRL4) ubiquitin E3 ligase core complex, which is composed of CUL4A or CUL4B, damage-specific DNA binding protein 1 (DDB1), and RING box protein 1 (RBX1, also called ROC1 or HRT1) [27]. The CRL4 ubiquitin E3 ligase core complex utilizes DCAFs (DDB1 and CRL4-associated factors), a set of WD40 domain-containing proteins, as the substrate-specific subunits to target specific protein substrates for proteolysis [26,113,114,115,116,117]. For example, the CRL4 core complex binds to a specific WD40 repeat-containing protein, CDT2 (also called DTL, L2DTL, or DCAF2), to target the DNA replication licensing protein CDT1 for DNA replication- or DNA damage-induced proteolysis through a PCNA-dependent process [115,116,118].

Since ubiquitin E3 ligases usually interact with their specific substrates [27], an ectopically expressed Flag-DNMT1 protein complex was biochemically isolated from cell lysates, and the DNMT1 protein complex-associated proteins were identified by means of mass-spectrometry-based proteome interrogation [108]. The mass-spectrometry-based analysis of the proteins associated with the isolated DNMT1 complex revealed that a methyl lysine reader protein, L3MBTL3, the component of CRL4 core complex, and a specific WD40 protein, DCAF5, were present in the DNMT1 complex [108]. Silencing of each individual component of CRL4 complexes, such as CUL4A or CUL4B, DDB1, or DCAF5, can block DNMT1 proteolysis induced by the loss of LSD1. The methylation-dependent proteolysis of DNMT1 can also be induced by the loss of the AKT1 kinase, which phosphorylates S143 to negatively regulate the SET7-mediated K142 methylation in DNMT1. Since ubiquitin E3 ligases usually directly interact with their protein substrates, it was found that conversion of K142 to alanine (K142A) abolishes the interaction between DNMT1 and CUL4A or CUL4B [108]. These studies suggest that the CRL4 ubiquitin E3 ligase complex employs DCAF5 as a substrate-specific subunit to target the methylated DNMT1 for ubiquitin-dependent degradation. The human *Dcaf5* gene is located within the chromosome 14q24.1–q24.4 locus, and deletion of this region is associated with congenital heart defects, brachydactyly, and mild intellectual disability [119]. However, since the DCAF5 protein does not contain a putative domain that binds to the methylated lysine residues, it remains unclear how CRL4 binds to the K142-methylated DNMT1 protein [116].

Human L3MBTL3 is a putative methyl-lysine reader because it contains the malignant brain tumor (MBT) domain with three MBT-repeated motifs. *L3mbtl3* mutations are found in medulloblastoma, and *L3mbtl3* is further implicated in prostate cancers and breast cancers [5,100,120]. The mouse *L3mbtl3* was initially identified as *Mbt-1*, and the homozygous deletion of the *Mbt-1* gene impairs embryonic blood development, leading to the developmental block at embryonic day 17.5–19.5 (E17.5–E19.5) [76]. The L3MBTL3 protein is considered a Polycomb Group (PcG) protein because it is isolated as a major component of the PRC1 complex that also contains SFMBT1, SFMBT2, LSD1, and CoREST [85,121]. L3MBTL3 has also been found to bind to the RBPJ-LSD1 complex involved in the Notch signaling pathway [122]. Although previous studies suggest that L3MBTL1 or L3MBTL2 mediates chromatin compaction or transcriptional repression by binding to mono- or dimethylated lysine residues in histones [5,123], it remains unclear how L3MBTL3 regulates embryonic development and other biological processes [5].

Since L3MBTL3 was present in the DNMT1 complex, it was confirmed that the endogenous DNMT1 and L3MBTL3 proteins indeed reciprocally interact with one another [108]. This interaction was further confirmed since the DNMT1-L3MBTL3 interaction was reduced when SET7 was knocked down. Conversely, expression of SET7 was found to enhance the interaction between L3MBL3 and DNMT1. However, the mutation of K142 to alanine (K142A) in DNMT1 abolished the interaction of DNMT1 with L3MBTL3, even in the presence of SET7 [108]. In addition, loss of PHF20L1, which binds to the methylated K142 in DNMT1, significantly promoted the DNMT1 and L3BTL3 interaction. Loss of L3MBTL3 also stabilized endogenous DNMT1 protein in LSD1 deficient cells. L3MBTL3 contains three MBT motifs, and investigation found that the second MBT motif is critical for L3MBTL3 binding to the mono-methylated K142 peptide, as L3MBTL3 binding to the methylated DNMT1 peptide is abolished when aspartate 381 in the second MBT repeat of L3MBTL3 is converted to asparagine (D381N) [5]. However, the in vitro binding of L3MBTL3 to the methylated K142 peptide is relatively weak, as the monomethylated lysine residue does not change the charge of the amino group of the original lysine residue. These studies together suggest that L3MBTL3 specifically recognizes the methylated K142 form of DNMT1 to target DNMT1 for proteolysis.

Since AKT1-mediated phosphorylation of serine 143 (S143) of DNMT1 negatively regulates the methylation of K142 by SET7 [111], loss of AKT1 promotes SET7-mediated K142 methylation of DNMT1. It was found that silencing of L3MBTL3 stabilized the DNMT1 protein in AKT1-deficient cells [108]. Furthermore, PHF20L1 was found to bind to the K142-methylated DNMT1 with its MBT/Tudor domain, and the loss of PHF20L1 releases the methylated K142 of DNMT1 [88]. Co-silencing of L3MBTL3 and PHF20L1 revealed that the loss of L3MBTL3 stabilized endogenous the DNMT1 protein in PHF20L1 deficient cells [108]. Since the loss of either L3MBTL3 or DCAF5 stabilized the DNMT1 protein in LSD1-deficient cells, these two endogenous proteins were found to be immuno-co-precipitated in the cell lysates [108]. Notably, since DNMT1 is required to maintain the DNA methylation patterns during DNA replication, it was found that the degradation of the methylated DNMT1 protein occurs in the S and G2 phases in the cell cycle [108].

## 5. E2F1 as a Substrate for L3MBTL3 and the CRL4^DCAF5^ Ubiquitin E3 Ligase Complex

The consensus K142 methylation degradation motif in DNMT1 is present in other non-histone proteins, including the methylation motif containing lysine 185 (K185) in E2F1 (Figure 1A), a critical transcription factor required for the expression of S-phase genes in the cell cycle. K185 has been shown to be methylated by SET7, and this methylation event induces E2F1 proteolysis [14]. It has been found that, while loss of LSD1 causes the proteolysis of E2F1 [14], co-silencing of L3MBTL3 or DCAF5 with LSD1 siRNAs leads to the stabilization of the E2F1 protein [108]. These studies indicate that L3MBTL3 and the CRL4^DCAF5^ ubiquitin E3 ligase complex also regulate the methylation-dependent proteolysis of the E2F1 protein.

## 6. SOX2 as a Substrate for L3MBTL3 and the CRL4^DCAF5^ Ubiquitin E3 Ligase Complex

It was found that the protein stability of SOX2 is also regulated by LSD1. Early studies found that the inhibition of LSD1 by chemical inhibitors led to the proteolytic degradation of SOX2, rendering the growth inhibition of embryonic stem cells and embryonic carcinoma cells [124,125]. It was found that human SOX2 contains a conserved lysine residue, K117 (equivalent to K119 of mouse SOX2), that can be mono-methylated by SET7 [67]. The methylated K117 is demethylated by LSD1 [109]. Loss of LSD1 leads to the proteolysis of SOX2 protein. The methylation of K117 by SET7 is prevented by AKT1-dependent phosphorylation on threonine 116 (T116) [67]. However, mutation of SOX2 to arginine (K117R) at K117 does not confer resistance to the loss of LSD1 [109], suggesting that SOX2 contains an additional methylated lysine residue. Indeed, it was found that lysine 42 (K42) of human SOX2 protein can also be methylated by SET7 to promote methylation-dependent proteolysis [109]. The methylation motif in K42 is R-V-K* (K* denotes the methylated lysine residue), which is different from the classical R-(T/S)-K or K-(T/S)-K methylation motif present in the methylation motifs of DNMT1, E2F1, and SOX2 K117 [14,67,111] (Figure 1A). However, it was found that the methylated K42 is also recognized by L3MBTL3 to target SOX2 for proteolysis [110], whereas the methylated K117 can also be weakly recognized by L3MBTL3. The regulation of SOX2 proteolysis by L3MBTL3 was also examined and confirmed by the use of UNC1215, a specific L3MBTL3 inhibitor [126]. Treatment of cells with UNC1215 blocks the degradation of SOX2 in LSD1-deficient cells [110]. Together, these findings indicate that the methylated SOX2 protein is targeted by the methyl lysine-binding protein, L3MBTL3, as well as the CRL4^DCAF5^ ubiquitin E3 ligase complex for ubiquitin-dependent proteolysis [110] (Figure 1B).

## 7. SMARCC1 and SMARCC2 as a Substrate for L3MBTL3 and the CRL4^DCAF5^ Ubiquitin E3 Ligase Complex

The targets of lysine methylation-mediated proteolysis can also be identified genetically, as loss of LSD1 can cause the degradation of methylated proteins [108,127]. One example is the recent identification of the mammalian SWI/SNF chromatin-remodeling complexes as a crucial target for lysine methylation-mediated proteolysis [127]. The mSWI/SNF (or BRG1/BRM-Associated Factor, BAF) complexes are large, ATP-dependent chromatin remodeling complexes assembled from multiple subunits or paralogues during development to regulate stem cells, differentiation, and cell fate determination [128,129,130]. The assembly of the mSWI/SNF complexes during embryonic development is regulated by changing subunits or incorporating paralogues to form various cell-/tissue-specific remodeling complexes for cell fate determination [129]. The mSWI/SNF complex assembly is biochemically initiated from the homo- or heterodimers of the core subunits, SMARCC1 (BAF155 or SRG3) and its paralogue SMARCC2 (BAF170), to incorporate SMARCD1 (or paralogues D2/D3), SMARCB1 (BAF47, INI1, or SNF5), and SMARCE1 (BAF57) to form the core complexes. The complexes subsequently incorporate ARID1 (ARID1A or ARID1B) or ARID2 to form different complexes. These complexes further assemble with the ATP catalytic subunits BRG1 (SMARCA4 or BAF190A), BRM (SMARCA2 or BAF190B), and PBRM1 (Polybromo-1 or BAF180) to form the canonical BAF, PBAF (polypromo-associated BAF), and ncBAF (non-canonical) complexes during development. In pluripotent mESCs, SMARCC1 and BRG1 form the esBAF complex, together with other subunits, including SMARCB1, ARID1A, and PBRM1. The esBAF complex is essential for the self-renewal and pluripotency of mESCs by interacting with critical pluripotent stem cell proteins OCT4, SOX2, and NANOG to regulate the ESC transcription circuitry [131,132,133,134,135]. It was found that expression of BRG1 and SMARCC1 can also enhance the formation of induced pluripotent stem cells (iPSCs) from mouse embryonic fibroblasts (MEFs) mediated by SOX2, OCT4, and Myc [135]. It is well established that the differentiation of mESCs leads to the assembly of various BRG1- and BRM-based mSWI/SNF complexes as additional subunits, such as BRM, a BRG1 paralogue, ARID1B, and SMARCC2, are incorporated into the complexes [133]. However, these new subunits are genetically different. For example, homozygous deletion of *Brg1* or *Smarcc1* causes early embryonic lethality in mice [136,137]. But the homozygous deletion of mouse *Brm* does not cause animal lethality, and *Smarcc2* null mutation is lethal after the mice are born [138]. Recent findings have shown that losses/mutations of mSWI/SNF genes are associated with more than 20% of human cancers and other human diseases [129,130]. The losses/mutations of a specific mSWI/SNF subunit usually predispose one a specific set of cancers through the formation of altered mSWI/SNF complexes. For example, loss of SMARCB1 or ARID1A causes proteolytic reduction in several other subunits, leading to the assembly of aberrant mSWI/SNF complexes for oncogenic processes [139,140,141,142]. While SMARCB1 mutations cause malignant rhabdoid tumors and epithelioid sarcoma, the dysfunction of ARID1A predisposes one to a wide variety of cancers, including ovarian clear cell carcinoma, endometrioid carcinoma, neuroblastoma, and bladder cancer. Alternatively, mutations of SMARCC1 are found in small cell lung cancer, whereas SMARCC2 is mutated in pancreatic cancer. Importantly, cancer cells containing the aberrant mSWI/SNF complexes with the mutation of a particular subunit are usually vulnerable to the loss of its remaining paralogue in the residual SWI/SNF complexes, generating the synthetic lethality of cancer cells [140,141,142,143,144]. For example, BRG1-mutated cancer cells are dependent on the presence of BRM, whereas ARID1A-mutated cancer cells are vulnerable to the deletion of ARID1B [140,141,142,143,144]. The mSWI/SNF proteins are most remarkably and primarily regulated by ubiquitin-dependent proteolysis [127,129,141,145], but until recently, very little has been known about the proteolytic mechanisms for the mSWI/SNF proteins.

Using the conditional floxed *Lsd1* mouse strain, it was found that homozygous loss of *Lsd1* alleles in the mouse brain leads to the disappearance of multiple subunits of mSWI/SNF chromatin remodeling complexes, including SMARCC1, SMARCC2, SMARCB1, BRG1, ARID1A, and PBRM1 proteins [127], associated with animal death immediately after the birth [127]. The disappearance of these mSWI/SNF proteins occurs due to their proteolysis by ubiquitin-dependent proteolysis. The mRNA levels of these mSWI/SNF subunits are not significantly affected by the loss of *Lsd1* alleles. It has been found that, since SMARCC1 and SMARCC2 are critical to initiating the assembly of the mSWI/SNF complexes [141,146,147,148], both the SMARCC1 and SMARCC2 proteins contain the lysine methylation motifs similar to those of DNMT1, SOX2, and E2F1 [127] (Figure 1A). Mutational analysis of the putative lysine methylation motifs in SMARCC1 protein indicates that lysine 482 (K482) and 615 (K615) of SMARCC1 are critical for lysine methylations by SET7 methyltransferase, and these methyl groups are dynamically removed by LSD1 demethylase. Since the esBAF complex only contains SMARCC1 in the mESCs and mouse F9 embryonic carcinoma cells [131,132,133,134], siRNA-mediated silencing of LSD1 led to the proteolysis of SMARCC1 and loss of self-renewal and pluripotency of mESCs or F9 cells, along with the disassembly of the esBAF complex and the loss of other esBAF proteins such as BRG1, ARID1A, and PBRM1 [127]. It was found that silencing of L3MBTL3 restored the SMARCC1 protein level, stabilized other esBAF proteins, rescued the esBAF complex, and restored the self-renewal and pluripotency of LSD1-deficient mESCs or F9 cells. These results indicate that L3MBTL3 regulates the proteolytic degradation of SMARCC1 in mESCs and F9 cells [110]. Expression of the SMARCC1 mutants that convert K482 and K615 to arginine (K482R and K615R) in mESCs and F9 cells is sufficient to confer the resistance of mESCs and F9 cells, as well as the assembly of the esBAF complex, to the loss of LSD1. Importantly, SOX2 and SMARCC1 interact in the ESCs and F9 cells. Although SOX2 is also sensitive to the loss of LSD1, the expression of SMARCC1 mutants caused an elevated level of the SOX2 protein, suggesting that the stabilized esBAF complex increases the stability and protein levels of SOX2. These results indicate that SMARCC1 is a critical target for LSD1 in the mESCs and F9 cells. Since the differentiation of mESCs leads to the expression of SMARCC2, both SMARCC1 and SMARCC2 are proteolyzed in mouse embryonic fibroblasts and various cancer cells. It was found that SMARCC2 contains a conserved lysine 457 (K457) with a methylation motif homologous to that of K482 in SMARCC1. The conversion of K457 to arginine (K457R) causes the resistance of SMARCC2 to the loss of LSD1, indicating that this is the critical lysine residue for SMARCC2 for lysine methylation-mediated proteolysis. Both SMARCC1 and SMARCC2 protein levels are relatively low in S-phase cells, suggesting that they are targeted for cell-cycle-regulated degradation similar to that of DNMT1. In *L3mbtl3* deletion mutant brain cells in mice, both SMARCC1 and SMARCC2 protein levels, as well as other methylated proteins, are elevated. These studies indicate that the lysine methylation-medicated proteolysis regulates the assembly and disassembly of the mSWI/SNF complexes by targeting the critical subunits, SMARCC1 and SMARCC2, for ubiquitin-dependent proteolysis. Further work is required in order to determine how the assembly and disassembly of the mSWI/SNF complexes are regulated in other stem cells and during development.

## 8. L3MBTL3 Inhibitor UNC1215 Serves as a PROTAC Molecule

The PROTAC technique was developed to target otherwise non-degradable proteins for ubiquitin-dependent proteolysis [149]. This technique links a chemical compound that binds to a ubiquitin E3 ligase to a chemical ligand that interacts with a target protein to mediate the target protein for proteolysis. Since UNC1215 binds to L3MBTL3 to block the degradation of methylated proteins, such as SOX2 [110], UNC1215 has been used to link to other chemicals that bind to other non-methylated proteins, such as FKBP12 and BRD2, for L3MBTL3-mediated ubiquitin-dependent proteolysis in the nucleus through the PROTAC mechanism [149].

## 9. Discussion

Lysine methylation-dependent proteolysis is a newly established research area, and only limited protein substrates have been characterized [108,109,110,127]. Many questions remain to be addressed. For example, it remains unclear how the methylated degron motif is defined and how the proteins involved specifically recognize the modified lysine residues in substrate proteins for proteolysis. Future work is also required in order to determine how methylation-dependent proteolysis is regulated in various cells and during development, as well as how these processes are altered in various diseases.

## Figures and Tables

**Figure 1 ijms-25-02248-f001:**
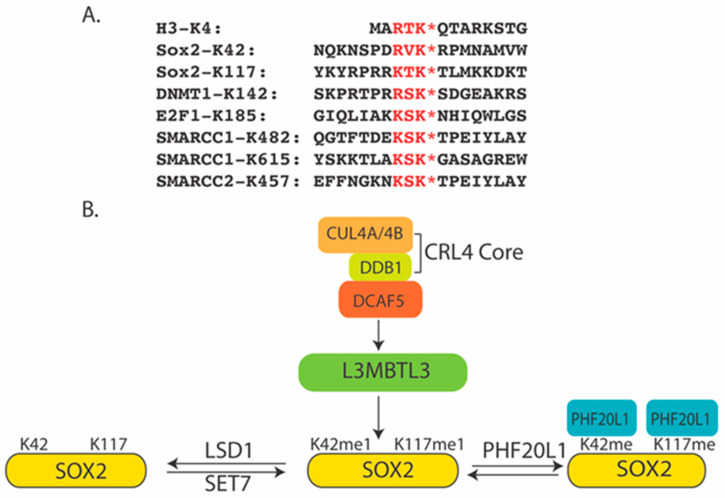
(**A**) The H3K4-like methylation motifs in methylated proteins. K* denotes the methylated lysine residues. (**B**) A model for the K42- and K117-methylation-dependent degradation of human SOX2 by L3MBTL3 and the CRL4-DCAF5 ubiquitin ligase complex. Other lysine methylated substrates in A are similarly regulated.

**Figure 2 ijms-25-02248-f002:**
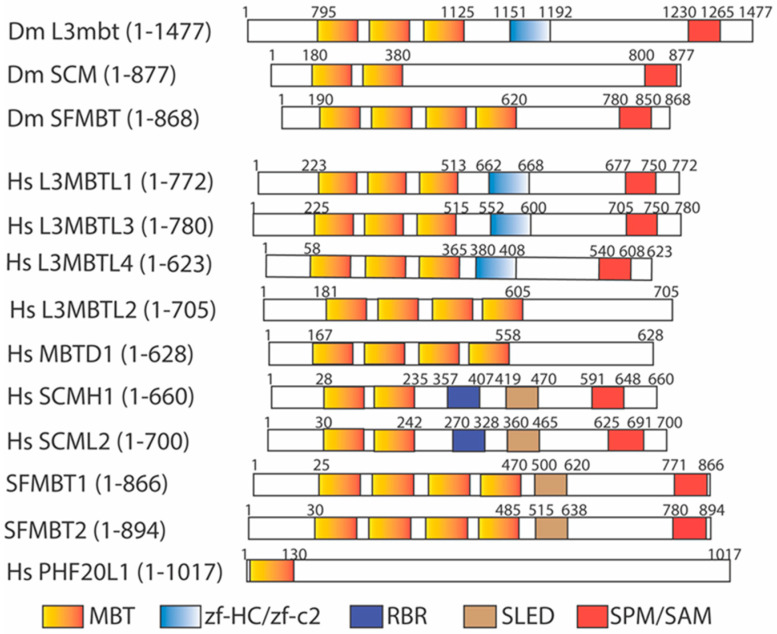
The domains in Drosophila (Dm) L3mbt, SCM, and SFMBT, and human (Hs) L3MBTL1, L3MBTL3, L3MBTL4, L3MBTL2, MBTD1, SCMH1, SCML2, SFMBT1, SFMBT2, and PHF20L1. MBT: the malignant brain tumor motif, zf-HC/zf-c2: zinc finger motif, RBR: RNA binding region, SLED: Scm-like embedded domain, SPM/SAM: SCM, PH, and MBT homology/Sterile α motif domain.

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
