# Peer review of "Lysine Methylation-Dependent Proteolysis by the Malignant Brain Tumor (MBT) Domain Proteins"

_ijms, 2024, doi:10.3390/ijms25042248_

Round 1

Reviewer 1 Report

Comments and Suggestions for Authors

Lysine methylation is an important post-translational modification that occurs in both histones and non-histone proteins. Recent studies have shown that methylated lysine residues in non-histone proteins can act as a signal for ubiquitin-dependent proteolysis. The SET7 (SETD7) methyltransferase enzyme transfers a methyl group from S-Adenosyl methionine to a specific lysine residue in a protein substrate, marking it for degradation through ubiquitin-dependent pathways. LSD1 (Kdm1a) acts as a demethylase, removing the methyl group from the modified protein. Methylated lysine residues are recognized by various proteins with specific reader domains. L3MBTL3, which contains the Malignant Brain Tumor domain, recognizes methylated lysine residues and recruits the CLR4DCAF5 ubiquitin ligase complex to target the methylated proteins for proteolysis. On the other hand, PHF20L1 recognizes methylated lysine residues to protect them from proteolysis. The lysine methylation-mediated proteolysis plays a crucial role in embryonic development, maintenance of pluripotency and self-renewal of embryonic stem cells, neural stem cells, hematopoietic stem cells, and other stem cell populations. It also regulates various biological processes. Dysregulation of lysine methylation-dependent proteolysis has been associated with different diseases, including cancers. Further characterization of lysine methylation and its role in regulating development and disease processes can provide valuable insights into these mechanisms. Understanding the precise mechanisms and functions of lysine methylation can potentially lead to the development of novel therapeutic approaches targeting these pathways in diseases. This is an interesting manucript, but I have several following concerns:

1. Abbreviations should be defined when they first appear in the text. Such as "CLR4", "PHF20L1", "STAT3",...

2. Lines 45 and 55, there is a mistake for the spelling of "NF-κB".

3. Please show the starting and ending amino acid positions of the protein domains in Figure 1.

4. Please improve the resolution of the Figures in the manuscript.

5. In this paper, the authors attempt to clarify the "Lysine Methylation dependent proteolysis by the Malignant Brain Tumor (MBT) domain Proteins", However, only a few Lysine methylation-related proteins (LSD1, SET7, and L3MBTL3) were involved in this study. It is well known that there are a large number of enzymes or proteins involved in the regulation of Lysine methylation. Why do the authors only introduce them in this manuscript?

6. Please unify the format of references in the article, including the author's name, the case of words in the title of the article, the writing of the name of the journal, and the page number.

Comments on the Quality of English Language

 Moderate editing of English language required.

Author Response

Dear Reviewer 1:

Thanks for excellent comments. Here are our ressponse:

1) Abbreviations should be defined when they first appear in the text.

Response: we thank the reviewer’s helpful comments and have defined the abbreviations when CRL4, DNMT1, and STAT3, first appear.

2) Line 45 and 55, the mistake in NF-kB.

Response: we thank the reviewer’s comments and sorry for the mistake. We have corrected the mistake.

3) Please show the starting and ending positions of the protein domains in Figure 1.

Response: Thanks. We have added the starting and ending positions of the protein domains.

4) Please improve the resolution of the Figures of the manuscript.

Response: we have modified the Figures to improve the resolution.

5) In this paper, the authors attempt to clarify the "Lysine Methylation dependent proteolysis by the Malignant Brain Tumor(MBT) domain Proteins", However, only a few Lysine methylation-related proteins (LSD1, SET7, and L3MBTL3) were involved in this study. It is well known that there are a large number of enzymes or proteins involved in the regulation of Lysine methylation. Why do the authors only introduce them in this manuscript?

Response: we thank the reviewer for the question. Although many histone proteins are lysine methylated, none of histone proteins is proteolyzed by lysine methylation. The lysine methylation-dependent proteolysis is a new concept that has only been characterized by the mentioned non-proteins, such as DNMT1, E2F, SOX2, SMARCC1, and SMARCC2.

6) Please unify the format of reference.

Response: we have used the EndNote and NIH format to unify the reference.

7) Moderate editing of the manuscript required.

Response: We have asked Dr. Rebecca Lim, an Asian American postdoctoral fellow who grew up in the US, to edit the revised manuscript.

Thank you!

Best regards,

Hui Zhang

Reviewer 2 Report

Comments and Suggestions for Authors

The paper named “ Lysine Methylation dependent proteolysis by the Malignant 2 Brain Tumor (MBT) domain Proteins” is a well done revision about the process of Lysine methylation and their implication in ubiquitin dependent proteolysis.

However, there are some things that the authors should check.

Although the title talks about MBT-dependent lysine methylation, only one section in the entire review is dedicated to this.

In figure 1 perhaps putting next to each group of proteins, the species could improve the understanding of the figure

In lines 229-230, authors talk about the mass spectrometry analysis of methyl lysine. It would be interesting if the technique used could be added (DDA...)

Comments on the Quality of English Language

English in fine and easy to understand, Minor editing of English language is required

Author Response

Dear Reviewer 2:

Thanks so much for the excellent comments. Here are our response:

1) Although the title talks about MBT-dependent lysine methylation, only one section in the entire review is dedicated to this.

Response: we thank the reviewer for the comment. The proteolysis of the MBT-dependent lysine methylation is a new field and only the research on L3MBTL3 regulated non-proteins, such as DNMT1, E2F, SOX2, SMARCC1, and SMARCC2, has been summarized. In response to the reviewer’s comment, we have moderately expanded the introduction of other MBT proteins on lysine methylation. However, none of these other MBT proteins have been described for lysine methylation-dependent proteolysis.

2) In figure 1 perhaps putting next to each group of proteins, the species could improve the understanding of the figure.

Response: we have modified Figure 1 to include the species of the proteins.

3) In lines 229-230, authors talk about the mass spectrometry analysis of methyl lysine. It would be interesting if the technique used could be added (DDA...)

Response: we thank the reviewer for the comments. We have included how L3MBTL3 and DCAF5 were identified by their association with the substrates such as DNMT1 and SOX2 using mass spectrometry analysis and cited the relevant references.

4) English in fine and easy to understand, Minor editing of English language is required.

Response: We have asked Dr. Rebecca Lim, an Asian American postdoctoral fellow who grew up in the US, to edit the revised manuscript.

Round 2

Reviewer 1 Report

Comments and Suggestions for Authors

The authors have addressed all my concerns. I recommend accepting it in current form.

Author Response

Dear Reviewers, thank you for your comments. We have made the changes in response to your comments 1-6. On comment 3, we are not sure whether our change is OK. Please let us know if there is a problem. Thank you so much!

Best,
Hui Zhang
